# Eliciting priors and relaxing the single causal variant assumption in colocalisation analyses

**Chris Wallace** *

Cambridge Institute for Therapeutic Immunology & Infectious Disease, and MRC Biostatistics Unit, University of Cambridge, Cambridge, United Kingdom

* cew54@cam.ac.uk

## Abstract

Horizontal integration of summary statistics from different GWAS traits can be used to evaluate evidence for their shared genetic causality. One popular method to do this is a Bayesian method, coloc, which is attractive in requiring only GWAS summary statistics and no linkage disequilibrium estimates and is now being used routinely to perform thousands of comparisons between traits. Here we show that while most users do not adjust default software values, misspecification of prior parameters can substantially alter posterior inference. We suggest data driven methods to derive sensible prior values, and demonstrate how sensitivity analysis can be used to assess robustness of posterior inference. The flexibility of coloc comes at the expense of an unrealistic assumption of a single causal variant per trait. This assumption can be relaxed by stepwise conditioning, but this requires external software and an LD matrix aligned to study alleles. We have now implemented conditioning within coloc, and propose a new alternative method, masking, that does not require LD and approximates conditioning when causal variants are independent. Importantly, masking can be used in combination with conditioning where allelically aligned LD estimates are available for only a single trait. We have implemented these developments in a new version of coloc which we hope will enable more informed choice of priors and overcome the restriction of the single causal variant assumptions in coloc analysis.

## Author summary

Determining whether two traits share a genetic cause can be helpful to identify mechanisms underlying genetically-influenced risk of disease or other traits. One method for doing this is "coloc", which updates prior knowledge about the chance of two traits sharing a causal variant with observed genetic association data in a Bayesian statistical framework. To do this using only summary genetic association data that is commonly shared, the method makes certain assumptions, in particular about the number of genetic causal variants that may underlie each measured trait in a genomic region. We walk through several data-driven approaches to summarise the prior knowledge required for this technique, and propose sensitivity analysis as a means of checking that inference is robust to uncertainty about that prior knowledge. We also show how the assumptions about

**Data Availability Statement:** Code to run all the simulations and analyses is available at https://github.com/chr1swallace/coloc-mask-paper. The updated coloc package may be installed as described at https://chr1swallace.github.io/coloc.

**Funding:** CW is supported by the Wellcome Trust https://wellcome.ac.uk/ (WT107881) and the MRC https://mrc.ukri.org/ (MC UU 00002/4). The funders had no role in study design, data collection and analysis, decision to publish, or preparation of the manuscript.

**Competing interests:** The authors have declared that no competing interests exist.

number of causal variants in a region may be relaxed, and that this improves inferential accuracy.

## Introduction

As genome-wide association studies (GWAS) have considered a greater diversity of traits in greater numbers of samples, comparative analyses of GWAS results have become a useful tool to explore the aetiological connections between different traits. For example, estimates of genetic correlation obtained via LD score regression quantify the average proportion of genetic variance of two traits that is shared across the genome, [1] although typically large sample sizes are required in both trait studies for accuracy. [2] Linking traits through genetics overcomes at least one major challenge of observational studies, reverse causality, and with careful design, can also address confounding. Epidemiologists have developed and widely deployed the technique of Mendelian randomization (MR), [3] which has been used, for example, to establish causal effects of factors such as alcohol intake on aspects of health. [4] The method uses a genetic variant or variants with established effects on one trait, and assesses whether a second trait is (proportionally) associated with these instrumental variables. Assuming certain assumptions hold true, [5] this provides evidence that the first trait is somehow causal for the second. While MR was originally envisaged as a test of causality of specific risk factors for which tests of causality might be confounded in observational studies, MR has been extended to routinely assess the potential for any GWAS trait to mediate another. [6] However, the ubiquity of genetic effects on some measurable aspect of human physiology or health, which have prompted suggestions of an omnigenic model, [7] raise concerns that LD between causal variants can violate the MR assumption that the instrumental variable is only associated with the outcome through the "mediating" trait. [8] This routine testing of all possible mediators is similar in design to the assessment of potential molecular causes of disease, which has been addressed through alternative approaches that focus not on whether one trait is causal for another, but whether two traits share the same causal variants in a single, LD-defined, genetic region, termed colocalisation.

While one such method is built on MR [9] and proceeds by filtering MR-positive associations via a test of heterogeneity in the estimated proportional effect across multiple SNPs in the region, another popular colocalisation method, coloc, [10] avoids MR assumptions altogether. Instead, coloc enumerates every possible configuration of causal variants for each of two traits, and calculates the support for that causal model in the form of a Bayes factor can be calculated under an assumption that at most one causal variant per trait exists in the region (see S1 Text). Each configuration corresponds to exactly one of five mutatally exclusive hypotheses about association and genetic sharing in the region:

$$H_0 \text{ : no association}$$
$$H_1 \text{ : association to trait 1 only}$$
$$H_2 \text{ : association to trait 2 only}$$
$$H_3 \text{ : association to both traits, distinct causal variants}$$
$$H_4 \text{ : association to both traits, shared causal variant}$$

The coloc approach has also been extended beyond pairs of traits, although computational efficiency scales poorly with numbers of traits [11, 12] unless decisions are binarised [13] and to

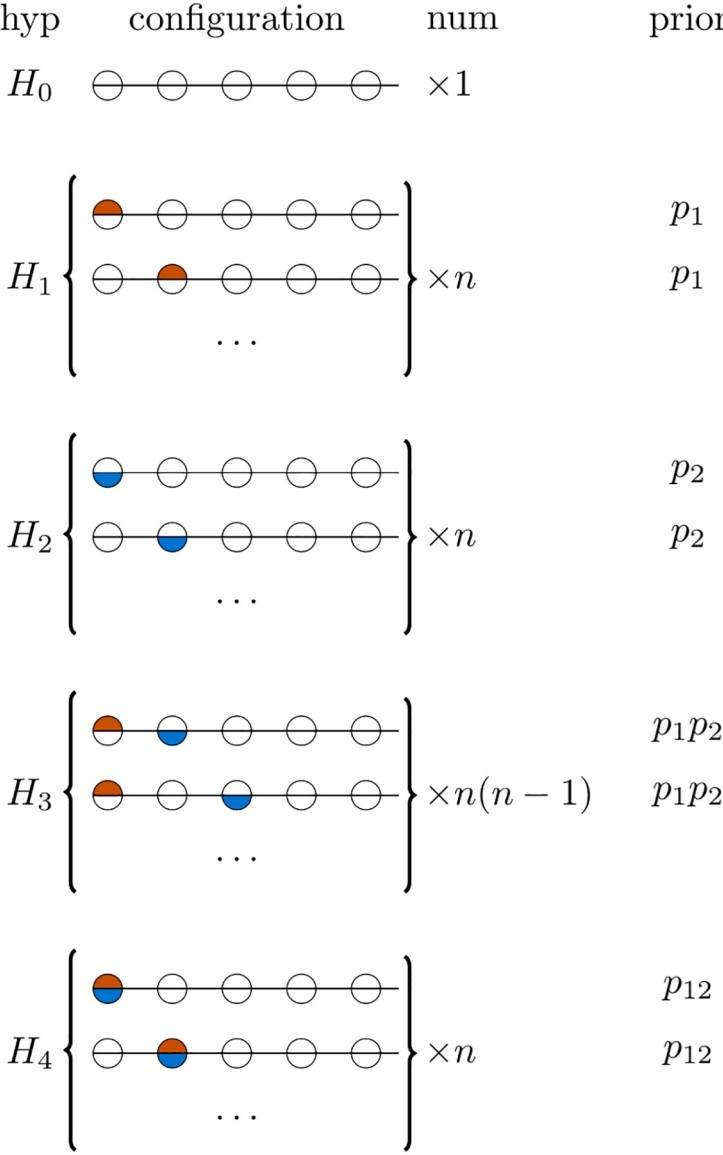

**Fig 1. Each hypothesis for coloc analysis $H_0 \ldots H_4$ may be enumerated by configurations, one configuration per row shown grouped by hypothesis.** Each circle in this figure represents one of $n$ genetic variants, and is shaded orange if causal for trait 1, blue if causal for trait 2. There are different numbers of configurations for each hypothesis, depending on the number of SNPs in a region, and the prior is set according to three prior probabilities so that all configurations within a hypothesis are equally likely.

deal with GWAS data that share controls, though at the expense of requiring raw genotype data [11].

As a Bayesian method, coloc requires specification of three informative prior probabilities: $p_1, p_2, p_{12}$ are, respectively, the prior probabilities that any random SNP in the region is associated with exactly trait 1, trait 2, or both traits (Fig 1). Although values for these were suggested in the initial proposal, [12] appropriate values should depend on specific datasets used, particularly for $p_{12}$, and no specific guidance on *how* this choice should be made was given.

One of the strengths of coloc is the simplicity of data required. The assumption of at most one causal variant per trait allows inference to be made through reconstructing joint models

across all SNPs from univariate (single SNP) GWAS summary data. [14, 15] Importantly, this requires no reference LD matrix and allows combining data from traits studied in differently structured populations. Further, p-values will suffice if internal or external estimates of minor allele frequency (MAF) are available, so that (unsigned) effect estimates and their standard errors can be re-constructed. However, the single causal variant assumption is convenient rather than realistic and when it does not hold colocalisation effectively tests whether the *strongest* signals for the two traits colocalise [10] which has been shown to be conservative [16].

e-CAVIAR [17] removes the assumption of a single causal variant per trait by integrating over the fine mapping posteriors for two traits, but requires signed effect estimates that are aligned to a reference LD matrix, that the traits are studied in the same population, and does not allow using any prior knowledge that shared causal variants are more or less likely than distinct variants. Perhaps the most challenging of these is the alignment of signed effect estimates to a reference LD matrix. This can be impossible in the case that signed estimates are not provided due to privacy concerns, [18] or that alleles are not provided. Even where alleles are available, palindromic SNPs (A/T, C/G) cannot be aligned unambiguously particularly for $MAF \approx 0.5$.

The assumption of a single causal variant in coloc may be relaxed by successively conditioning on the most significant variants for each trait, and testing for colocalisation between each pair of conditioned signals, although this requires either complete genotype data or use of external software such as CoJo [19] together with signed and LD-aligned effect estimates to allow reconstruction of conditional regression effect estimates.

To support more accurate coloc analyses, we explored a variety of data-driven approaches to inform prior choice across a range of traits and developed a framework to explore sensitivity of conclusions to the priors used. Further, we implemented an existing conditioning approach in the coloc package, but also developed an alternative approach to conditioning which does not require aligned LD and effect estimates, to offer an option to deal with multiple causal variants which preserves the simplicity of the data required for coloc analyses.

## Results

We used Scopus to identify 60 papers which cited coloc [10] and were published in 2018. Out of these, we extracted the subset of 25 papers that were both applied papers (rather than methodological) and for which full text could be accessed (S1 Table). The studies covered a variety of trait pairs, generally integrating a disease GWAS with molecular quantitative trait loci (QTL) data, [20–39] but also comparing pairs of disease GWAS, [40] eQTL and pQTL [41, 42] or eQTL and other molecular traits. [43, 44] Only four studies considered the potential for multiple causal variants in a region, either discussing the implications on their results, or using conditioning in at least one trait, and 22 out of 25 studies used the software default priors across this diverse range of trait pairs.

Given that it is likely that the prior probability of colocalisation may depend on the trait pairs under consideration, we decided to evaluate the effect of mis-specifying prior parameters and/or not conditioning when multiple causal variants exist.

### The importance and elicitation of prior parameter values

Before examining the robustness of inference to changes in prior values, we elucidate some properties of prior parameters. While priors are expressed per SNP, our hypotheses and posterior relate to a region—a set of $n$ neighbouring SNPs. The prior that one SNP in the region is causally associated with trait 1 is $\approx np_1$ (and similarly $np_2$ for trait 2, $np_{12}$ for colocalisation). All these scale with the number of SNPs—the larger the set of SNPs we consider, the greater

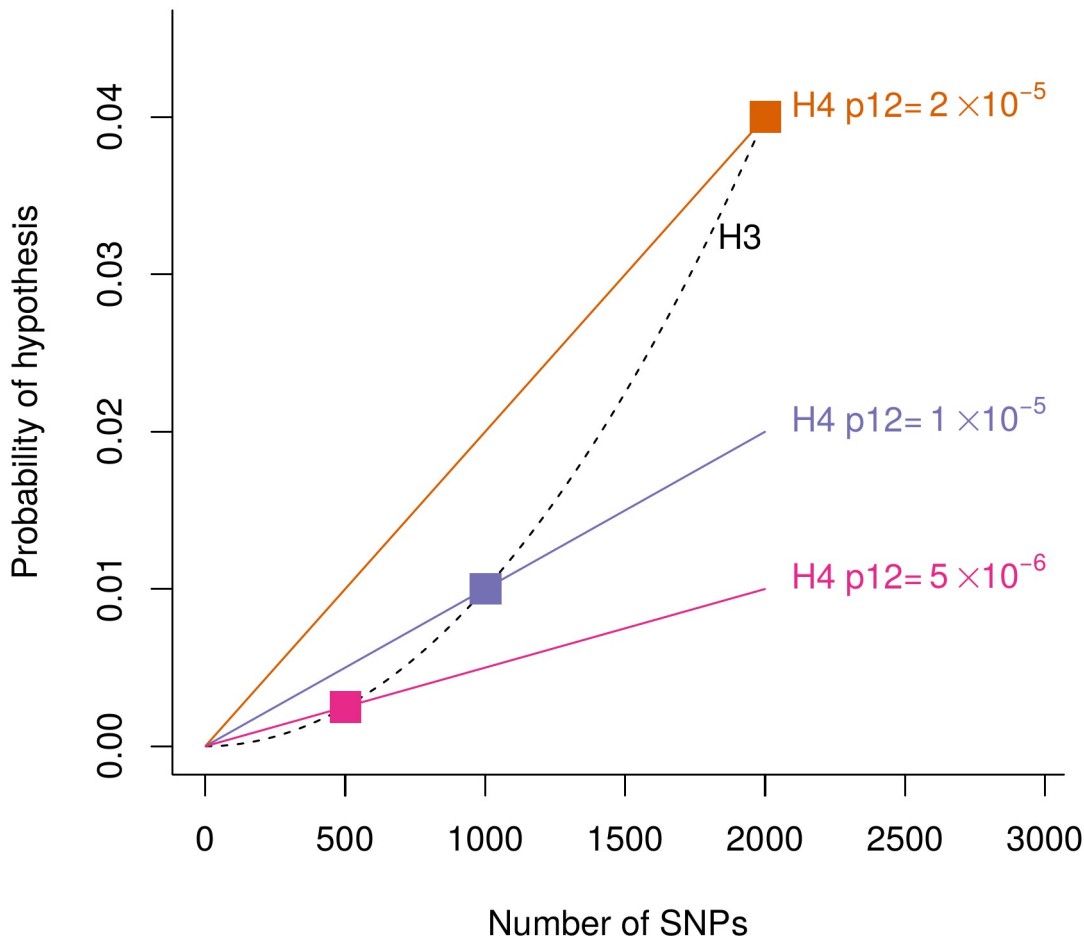

**Fig 2. Effects of varying $p_{12}$ on the prior for $H_4$ (coloured lines) compared to $H_3$ (dashed line) as a function of the number of SNPs in the region.** For all plots $p1 = p2 = 10^{-4}$ is constant. The coloured squares highlight points $P(H_3) = P(H_4)$ for different $p_{12}$.

the chance one of them is causal for any trait. Despite this, the prior odds for $H_4/H_1$—colocalisation compared to association of trait 1 only—remains constant at $p_{12}/p_1$.

The prior for $H_3$ (two distinct variants for the two traits) is $\approx n(n-1)p_1 p_2$ which scales with the square of $n$. This means that prior odds of the two hypotheses of greatest interest, $H_4/H_3$, depends not only on the per SNP prior of causality for one or other trait, but also on the number of SNPs in a region, to the extent that the same $p_1, p_2, p_{12}$ may favour either $H_3$ or $H_4$ as larger regions are considered (Fig 2). This effect can be understood by noting that both $H_3$ and $H_4$ imply that each trait has exactly one causal variant in the region. Simple combinatorics implies that as the number of SNPs in a region increases, then the number of ways two different SNPs can be causal for the two traits ($H_3$) increases more rapidly than the number of ways one SNP can be causal for both ($H_4$). Hence, $H_3$ becomes relatively more likely than $H_4$ as the number of SNPs in the region increases.

**Marginal priors.**   To elicit values for $p_1, p_2$, we reparameterise, focusing on the possible marginal events for any SNP:

$A_1$ :   SNP is causally associated to trait 1   Prob $q_1 = p_1 + p_{12}$

$A_2$ :   SNP is causally associated to trait 2   Prob $q_2 = p_2 + p_{12}$

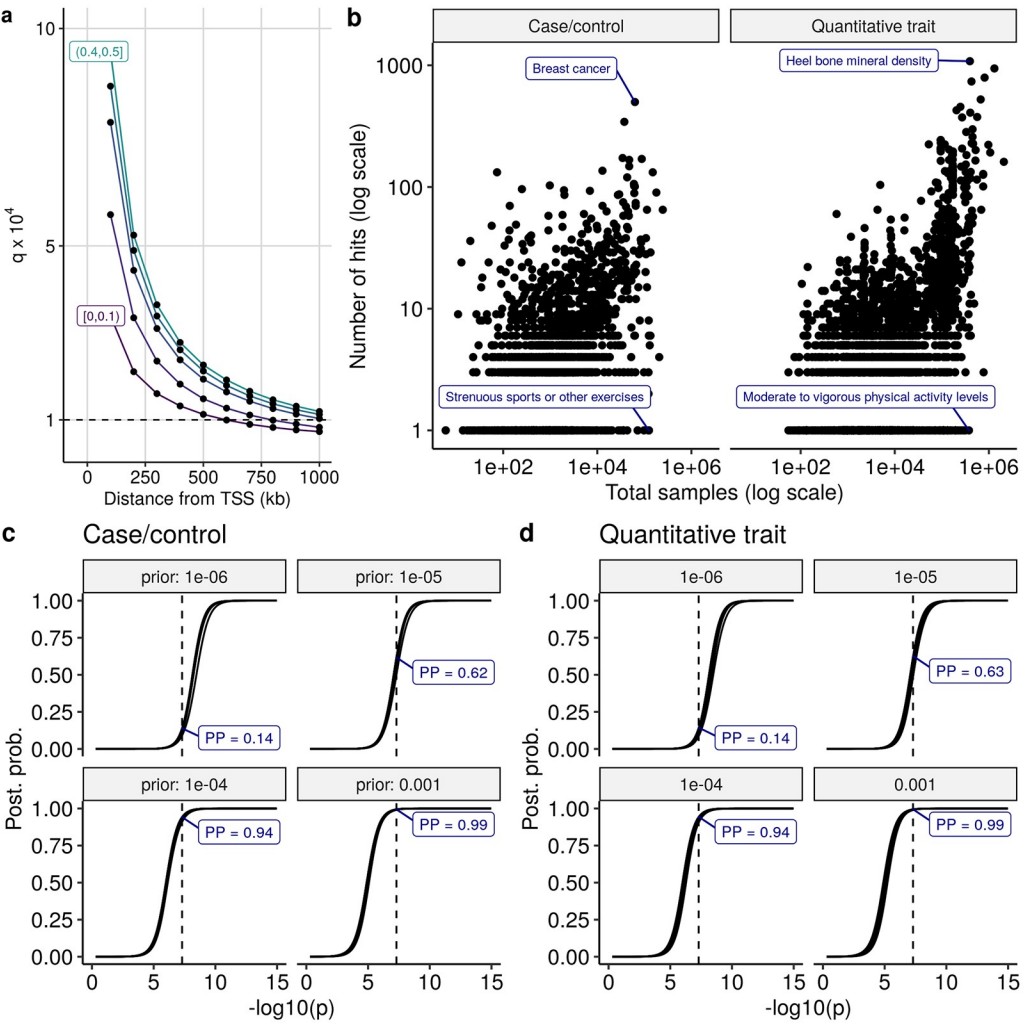

**Fig 3. Determining plausible priors $q_1$, $q_2$. a** $q$. estimated for eQTLs as the ratio of estimated number of LD-independent significant eQTL variants divided by number of SNPs considered for an eQTL analysis in GTeX whole blood samples in successively larger windows around a gene TSS. Separate lines show findings in 5 equal groups of MAF, with the top and bottom groups labelled. **b** The number of hits claimed per study according to the GWAS catalog. $q$. could be estimated as number of hits / number of common SNPs ($\sim$ 2, 000, 000). **c** Posterior probability of association at a single SNP as a function of -log10 p values for varying values of $q$. We considered both case/control and quantitative trait designs, and a range of MAF (0.05-0.5) and sample size (2000,5000,10000). The relationship between -log10 p (x axis) and posterior probability of association (y axis) is consistent across all designs, affected only by the prior probability of association ($q_1$, $q_2$). The vertical line indicates $p = 5 \times 10^{-8}$, the conventional genome-wide significance threshold in European populations.

Note that in this notation, $A_1$ and $A_2$ are not mutually exclusive, so that colocalisation is $A_1 \cap A_2$. $q_1$, $q_2$ can be estimated empirically by considering evidence from the wealth of single trait association data that already exists. For eQTLs, we use GTeX data [45] and find that $q$. is dependent on the MAF of SNPs considered, which reflects variable power with fewer true eQTL variants detectable at lower MAF, and search window around the gene considered as previously noted, tending to $10^{-4}$ for common SNPs and windows $\sim 1$ mb (Fig 3).

The GWAS Catalog [46] enables us to consider something similar by aggregating over 5000 GWAS studies. We find, as expected, and again as previously noted,[47] that the number of hits per study increases steadily with increasing sample size (Fig 3), but that the count also

depends on the class of trait considered, with "harder" endpoints such as breast cancer and heel bone mineral density identifying orders of magnitude more associations compared to "weaker" endpoints such as tendency to strenuous sports or activity levels. The largest studies find $\sim$ 100–1000 hits out of $\sim$ 2 million common SNPs leading to estimates that 5 in 10,000–100,000 common SNPs are detectably causal for these traits which corresponds to $q_{.} \in [5 \times 10^{-5}, 5 \times 10^{-4}]$. Even with the largest studies, these estimates must be considered likely to continue to increase with sample size, and therefore conservative. Using conservative priors for $p_1, p_2$ in colocalisation analysis is likely to reduce power to detect either shared or distinct causal variants, because weaker signals may be wrongly interpreted as trait-unique or null. However, estimates from the largest available studies also represent at upper bound on the proportion of variants likely to be *detectably* associated in any new study from the same class of traits, and therefore relaxing the priors further might result in over-stating the evidence for causal variants and erring towards false detection of shared or distinct causal variants.

An alternative approach is to choose the prior according to the p-value that we would consider significant. The threshold of $p < 5 \times 10^{-8}$ has been widely adopted as "genome-wide significant" for GWAS studies in European populations. Across a range of designs (case/control or quantitative trait, with varying MAF and sample size), we see that a prior of $q_{.} = 10^{-4}$ gives a strong posterior probability of association ($\approx 0.94$).

The default coloc marginal prior of $q_1 = q_2 = 10^{-4} + p_{12} \approx 10^{-4}$ is thus supported by the convergence of these three approaches to values of the order of $10^{-4}$.

**Prior probability of joint or conditional causality.** $q_1$ and $q_2$ themselves place some constraints on $p_{12}$. On the one hand, the chance of joint causality cannot be greater than the chance of causal association with either trait. On the other hand, if traits were independent, then causal variants for each trait would happen to co-occur at the same location with probability $q_1 \times q_2$. However, simulations show that the distribution of expected posterior probabilities vary considerably with $p_{12}$ over this range (Fig 4), indicating that we need to make some effort to elicit plausible values. The results suggest that the coloc default of $p_{12=} = 10^{-5}$ may be overly liberal, with data simulated under $H_3$ having posterior support for $H_4$, particularly for smaller samples, and that $p_{12} = 5 \times 10^{-6}$ may be a more generally robust choice.

We consider different approaches to determine data-driven estimation of $p_{12}$. First, we can set a lower bound if we take into account that not all of the genome is understood to be functional. Estimates of the functional proportion vary considerably, from 25% [48]–80%. [49] Even for traits that are genetically independent, knowing that a SNP is causal for one trait implies it is functional, and thus more likely to be causal for another trait then a random SNP that may or may not be functional. Assuming the proportion of genetic variants that are functional is $f$, the probability of co-occurence by chance alone is $q_1 q_2/f$ (see S1 Text).

In the case of comparing two GWAS studies, it may be possible to estimate the genetic correlation, $r_g$. We show in S1 Text that, when shared variants do not have any systematically different distribution of allele frequencies or effects compared to non-shared variants,

$$|r_g| \le \frac{n_{12}}{\sqrt{(n_{12} + n_1)(n_{12} + n_2)}} = \frac{p_{12}}{\sqrt{q_1 q_2}}$$

where $n_{12}, n_1, n_2$ are the number of variants shared, distinct to trait 1 and distinct to trait 2.

Putting these together, we find

$$\frac{q_1 q_2}{f} < p_{12}, \quad |r_g|\sqrt{q_1 q_2} < p_{12}, \quad p_{12} < \min(q_1, q_2).$$

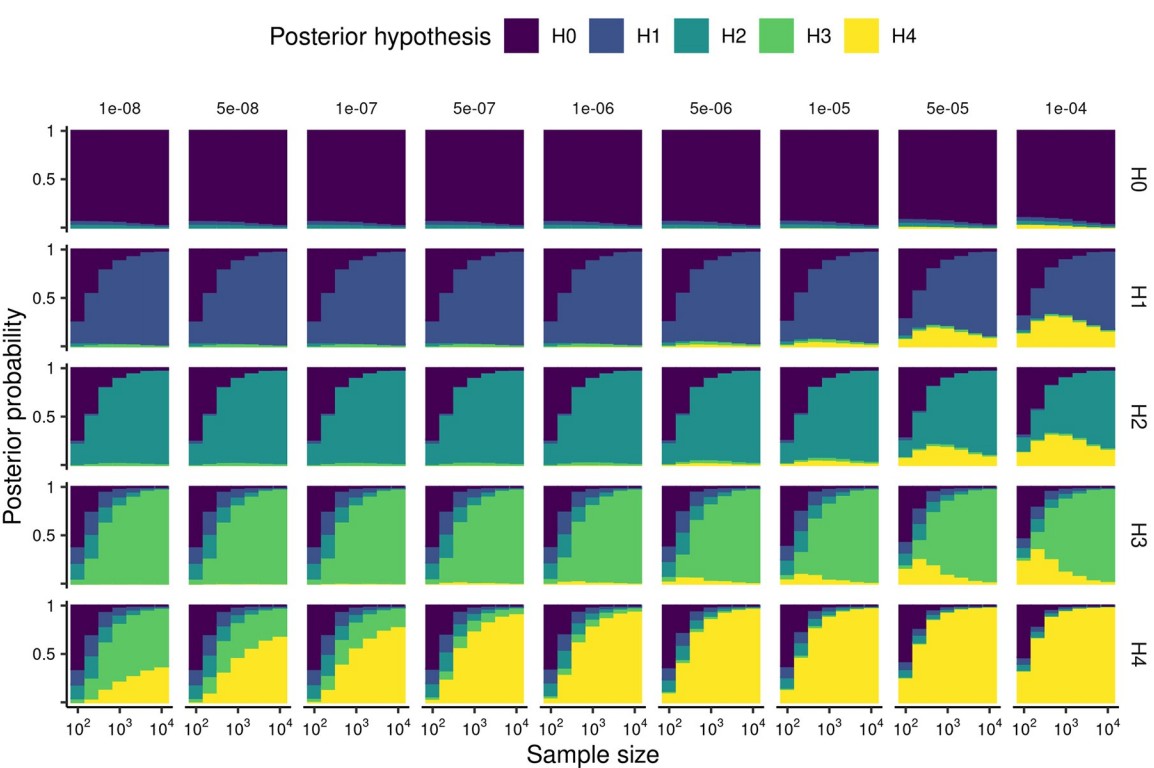

**Fig 4. Distribution of expected posterior probabilities across a wide range of simulated data.** In all analyses we fixed $p_2 = p_1 = 10^{-4}$ and varied $p_{12}$. Coloured bar heights represent the average posterior probability for each hypothesis over the set of simulations for a given simulated hypothesis and sample size.

Second, where studies of both traits are well powered, then methods for joint analysis of trait pairs may be informative. For example, gwas-pw [50] extends the original coloc by using empirical Bayes to estimate per-hypothesis priors via joint analysis of all regions genomewide. However, this comes at a cost of ignoring the dependence of per-hypothesis priors on the number of SNPs in a region, and even in simulated data did not generate consistent estimates. This latter may reflect the limited information that exists in any pair of GWAS (the number of regions where detectable signals exist for both traits). Nonetheless, such an approach can probably give a useful order of magnitude estimate for $p_{12}$.

Finally, in the absence of data about joint trait association at the genome-wide level, it is necessary to rely more on investigator judgement, and here it may helpful to consider conditional probabilities

$$p_{12} = P(A_1 \cap A_2) = P(A_1|A_2) \times P(A_2) = q_{1|2} \times q_2$$

The term $q_{1|2}$ represents the probability that a SNP, already known to be causal for trait 2, is also causal for trait 1. In asymmetric analysis such as GWAS and eQTL, it may be simpler to condition on one event rather than the other—does the investigator have a clearer idea of the chance that a SNP that causally regulates gene expression in a given tissue is causally associated with a disease or the chance that a SNP that is causally associated with a disease does so via transcriptional regulation in that same tissue?

To aid translation of priors between the two parameterisations discussed here, we have created an online tool "coloc explorer" at https://chr1swallace.shinyapps.io/coloc-priors.

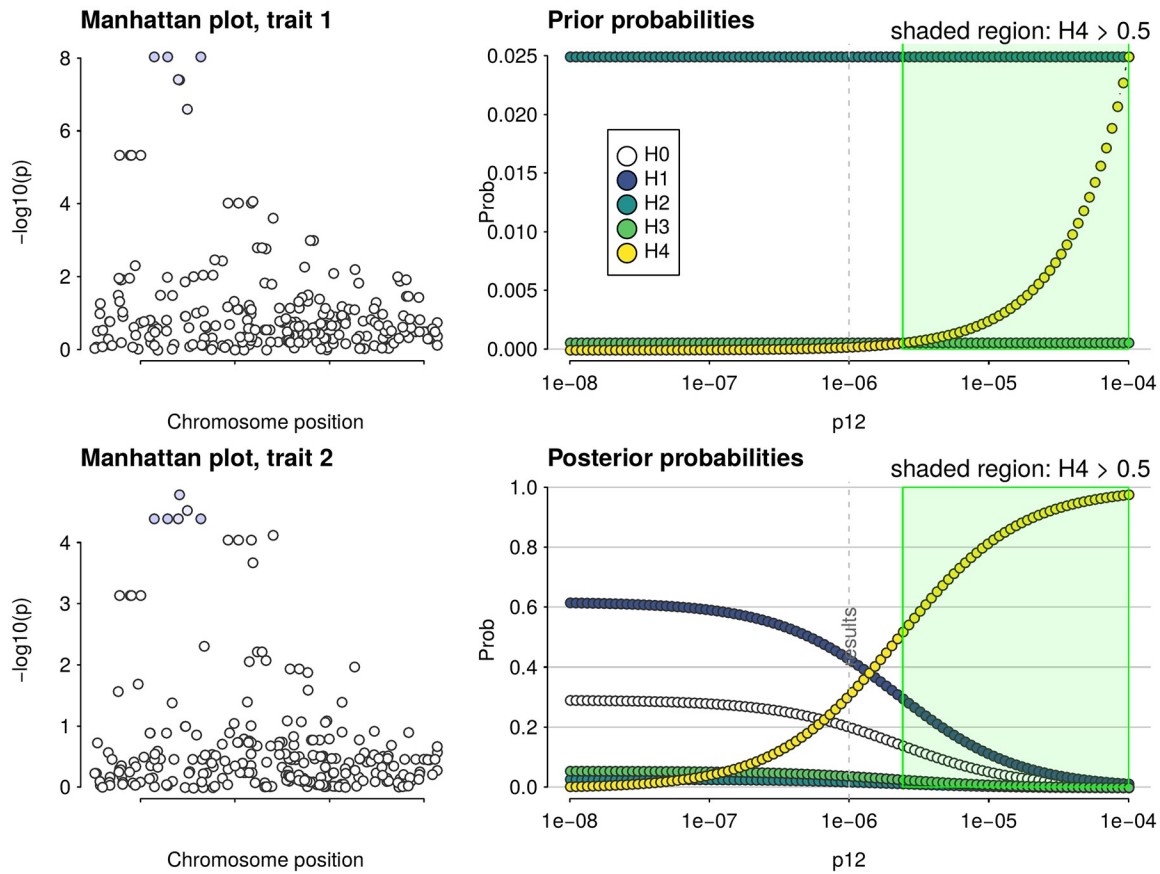

**Fig 5. Example of sensitivity analysis on a dataset which shows evidence for colocalisation at a predefined rule of posterior $P(H4)$ > 0.5 only when the prior beliefs in H3 and H4 are approximately equal.** The left hand panels show local Manhattan plots for the two traits, while the right hand panels show prior and posterior probabilities for H0-H4 as a function of $p_{12}$. The dashed vertical line indicates the value of $p_{12}$ used in initial analysis (the value about which sensitivity is to be checked). $H_0$ is omitted from the prior plot to enable the relative difference for the other hypotheses to be seen.

**Sensitivity analysis.** In the expected case that an investigator does not have a strong prior belief in a single value for $p_{12}$ we can use sensitivity analysis to consider whether conclusions are robust over a range of plausible values. Helpfully, it is not necessary to reanalyse the complete dataset multiple times. Given that

$$P(H_i|D, \pi) \propto BF_i \times P(H_i|\pi) = \frac{P(D|H_i)}{P(D|H_0)} \times P(H_i|\pi)$$

where $D$ represents study data and $\pi = (p_1, p_2, p_{12})$ is the prior parameter vector used for analysis, we can derive posterior probabilities under an alternative prior parameter $\pi^*$ as

$$P(H_i|D, \pi^*) \propto P(H_i|D, \pi) \times \frac{P(H_i|\pi^*)}{P(H_i|\pi}$$

and so we can rapidly explore sensitivity of inference to changes to $p_{12}$. Fig 5 shows an example where conclusions depend heavily on the relative prior belief in $H_3$ and $H_4$ and a conclusion of colocalisation by a decision rule of $P(H_4|D, \pi) > 0.5$ is only valid if prior beliefs are that $H_4$ is at least as likely as $H_3$. An alternative example where results are robust over a wide range of $p_{12}$

is shown in S1 Fig. Detailed instructions to run a sensitivity analysis are given at http://chr1swallace.github.io/coloc/articles/a04_sensitivity.html.

## Conditioning and masking to allow for multiple causal variants

In order to deal with multiple causal variants in a region, we implemented the CoJo approach [19] within the coloc package. We also propose an alternative to conditioning which does not depend on allelic alignment and can be used with p-values alone: masking. Stepwise regression proceeds by identifying the top SNP, and then re-estimating association statistics across all other SNPs to test whether they provide any additional information to infer the trait of interest. Conditional effect estimates at SNPs in LD with the top SNP(s) differ from their unconditional values, so that they capture the residual evidence for association, but conditional and unconditional effect estimates are (effectively) the same at SNPs independent from the top SNP(s). Our proposed masking algorithm relaxes the assumption of a single causal variant by instead assuming that if multiple causal variants exist for any individual trait, they are in linkage equilibrium. It therefore first identifies lead SNPs, then successively masks all SNPs in LD with the top signals(s), testing for significant association in the remainder, and adding SNPs sequentially while residual association remains (Fig 6). When colocalising, each lead SNP is taken in turn, and any SNPs in LD with *any other* lead SNP are masked, by setting the per-SNP Bayes factor to 1 for any SNP-specific hypothesis relating to that SNP/trait pair. We have implemented both approaches in the development version of the coloc package, https://github.com/chr1swallace/coloc/tree/condmask, and document their use at http://chr1swallace.github.io/coloc/articles/a05_conditioning.html.

We compared conditioning and masking to single coloc analysis across a variety of simulated datasets (Figs 7 and 8). A single coloc comparison generally relates to the strongest signals for each of the two traits, as previously reported, [10] which can miss colocalising signals that are secondary to a primary independent signal (Fig 7, row 3) or that have differently ordered effect sizes (Fig 8, row 5). Conditioning allows more distinct comparisons and shows a marked improvement on single coloc, in particular being able to identify a greater proportion of the truly colocalising signals. Masking increases the number of comparisons compared to single coloc, but is less informative than conditioning. In particular, the number of comparisons that cannot be clearly assigned to a specific causal variant pair (at least one lead SNP does not have $r^2 > 0.8$ with a causal variant) increases when multiple causal variants are in LD (S2 and S3 Figs) and this fraction of comparisons are often inaccurate, finding posterior support for $H_3$ when $H_4$ is true.

## Discussion

This paper has focused on two practical aspects of Bayesian colocalisation analysis that hitherto have not received detailed attention. The ability of Bayesian methods to incorporate prior knowledge and beliefs is a strength of the coloc approach, but also places onus on a researcher to evaluate their prior beliefs. Elicitation of informative priors is a subject that has received much attention in the statistical literature [51] but rather less within the genetics community. Nonetheless, the use of Bayesian methods in genomics is growing in popularity, as a natural way to fit joint models to large and complex data sets and to enable integrative analysis over different traits or datasets. When data are large, and the number of events is also large, then empirical Bayes can enable an analyst to learn the prior from the same data used for testing. However, in the case of smaller studies or less common events, the wealth of existing information from other large studies as well as investigators' own beliefs can be used.

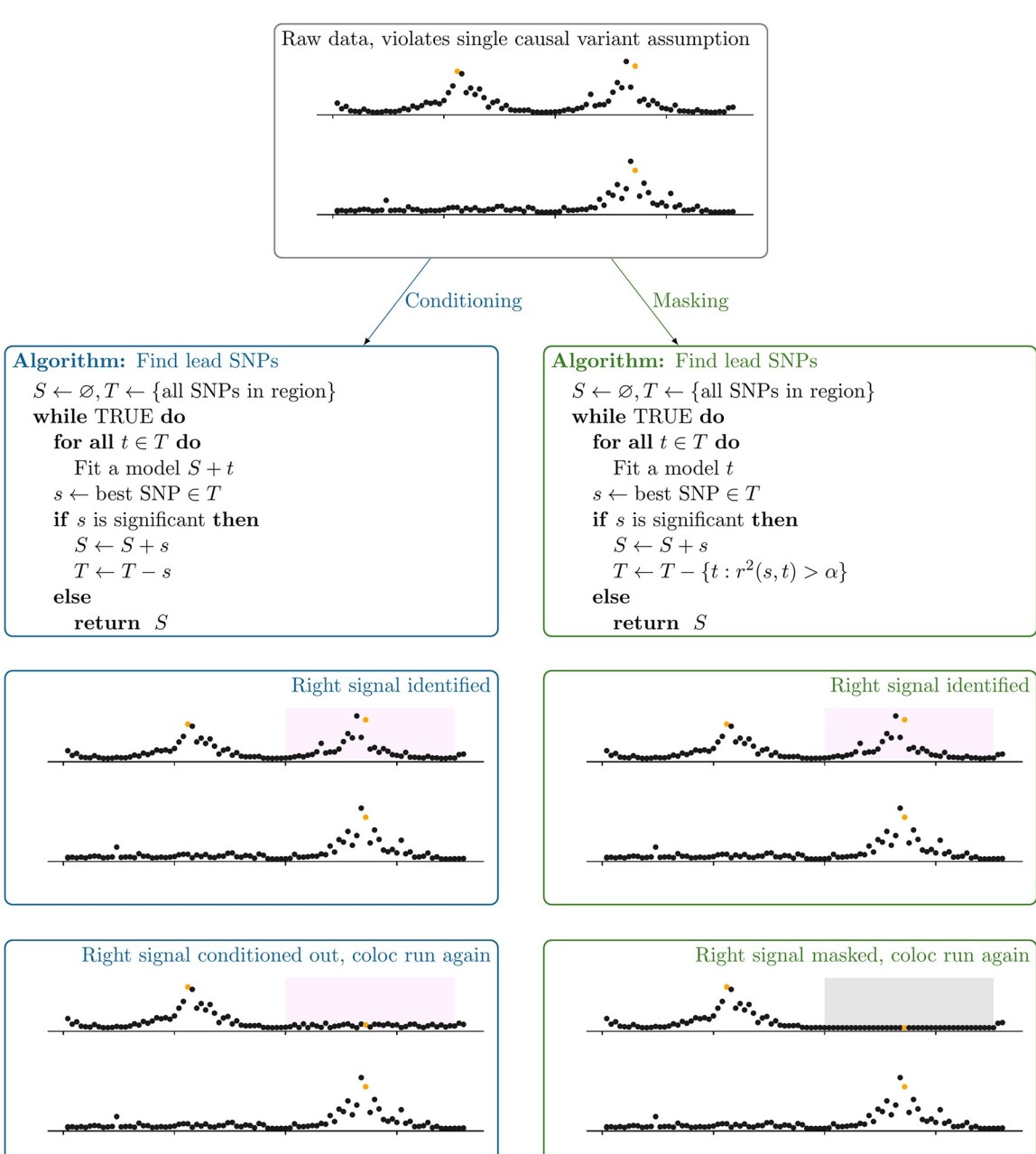

**Fig 6. Masking as an alternative strategy to conditioning when attempting to colocalise trait signals with multiple causal variants in a region.** Top panel: input local Manhattan plots, with causal variants for each trait highlighted in red. We can use conditioning (left column) to perform multiple colocalisation analyses in a region. First, lead SNPs for each signal are identified through successively conditioning on selected SNPs and adding the most significant SNP out of the remainder, until some significance threshold is no longer reached. Then we condition on all but one lead SNP for each parallel coloc analysis. Note that when multiple lead SNPs are identified for each trait, eg $n$ and $m$ for traits 1 and 2 respectively, then $n \times m$ coloc analyses are performed. When an allele-aligned LD matrix is not available, an alternative is masking (right column) which differs by successively restricting the search space to SNPs not in LD with any lead SNPs instead of conditioning. Multiple coloc analyses are again performed, but setting the per SNP Bayes factor to 1 for hypotheses containing SNPs in LD with any but one of the lead SNPs. Note that for convenience of display, all SNPs in $r^2 > \alpha$ with the lead SNP are assumed to be in a contiguous block, shaded gray.

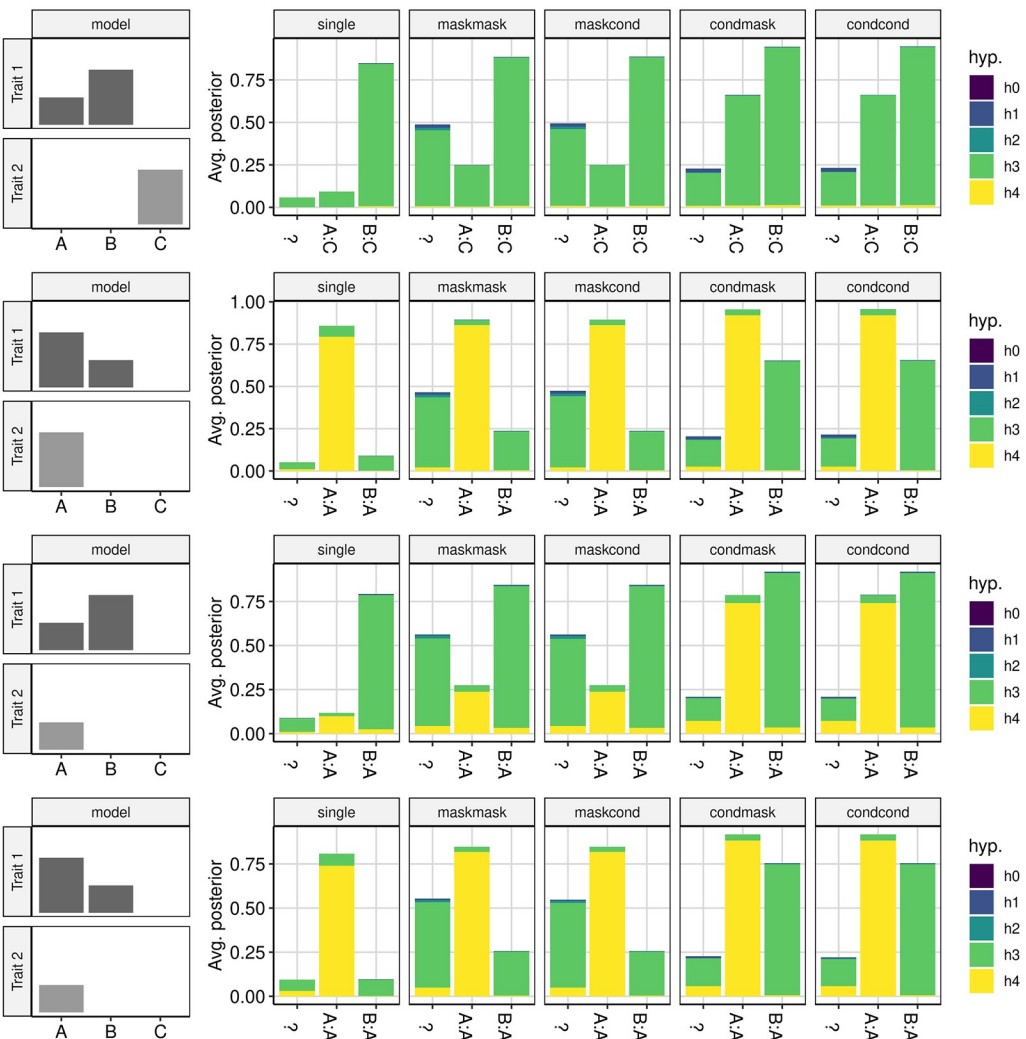

**Fig 7. Average posterior probabilities for each hypothesis under different analysis strategies when trait 1 has two causal variants, A and B, and trait 2 has just one.** The left column shows the identity of causal variants for each trait and their relative effect sizes under four different models. The right column shows the average posterior that can be assigned to specific comparisons for of variants for trait 1: trait 2. We exploit our knowledge of the identity of the causal variants in simulated data to label each comparison according to LD between the lead SNP for each trait and the simulated causal variants. When labels cannot be unambiguously assigned ($r^2 < 0.8$ with any causal variant) we use "?".

For coloc, the choice of marginal prior parameter values can be readily informed in this way. For joint causality this is harder and while we suggest and walk through several alternative ways of doing this the conclusions we draw are not universally applicable; each investigator should use both available data and their own judgement to elicit their own prior beliefs and those of their co investigators. Perhaps the most widely applicable are the results of simulations, that suggest values of the order $p_{12} \approx 5 \times 10^{-6}$ lead to robust inference over a range of scenarios, but the adoption of sensitivity analysis will help evalutate robustness of inference to changes in prior parameter values.

Attempts to colocalise disease and eQTL signals have ranged from underwhelming [52] to positive.[53] One key difference between outcomes is the disease-specific relevance of the cell types considered, which is consistent with variable chromatin state enrichment in different

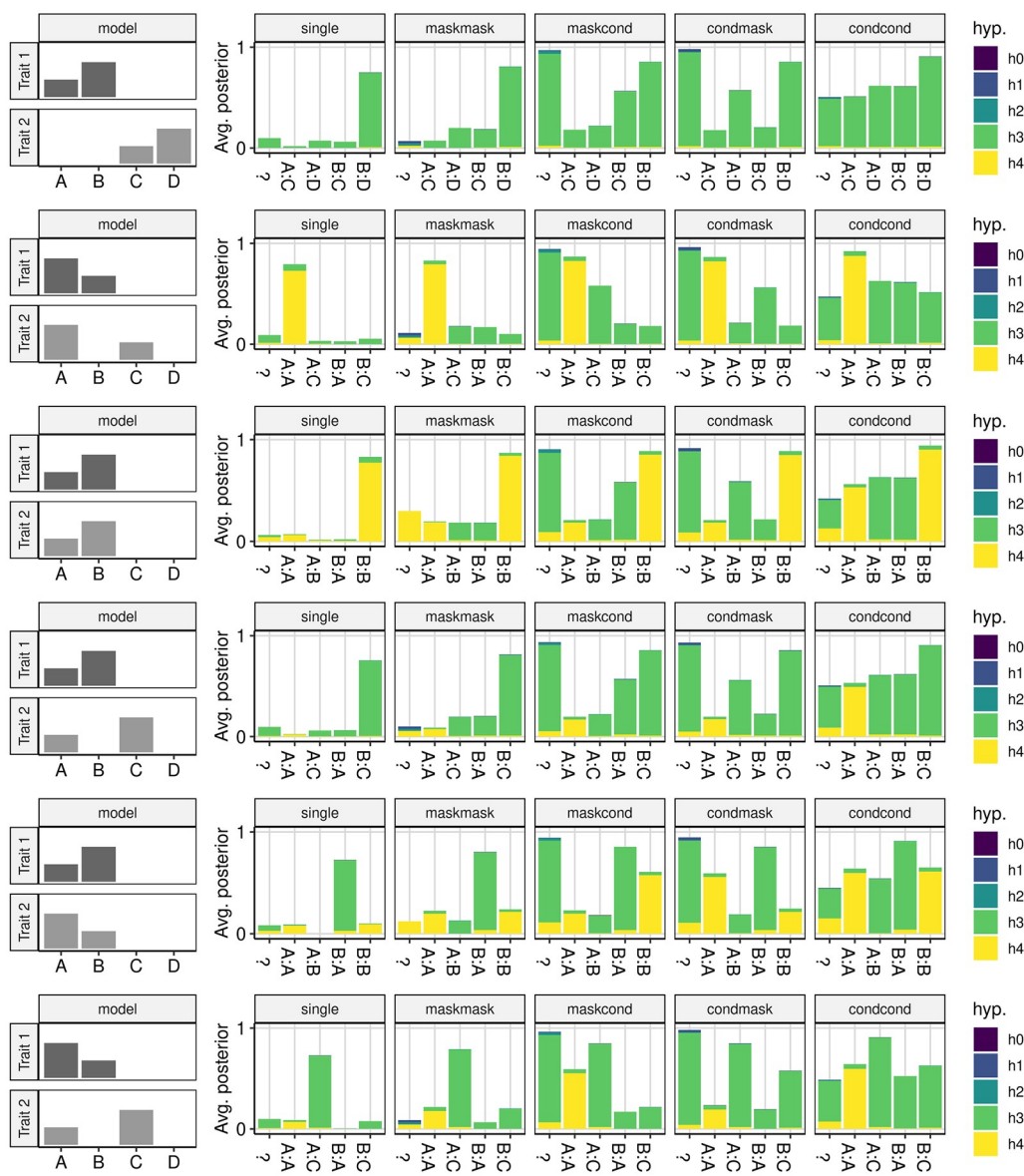

**Fig 8. Average posterior probabilities for each hypothesis under different analysis strategies when both traits have two causal variants.** Information is displayed as described in Fig 7.

GWAS according to cell type.[54] For example, studies considering the overlap of open chromatin and GWAS signals have convincingly shown that tissue relevance varies by up to 10 fold, [55] with pancreatic islets of greatest relevance for traits like insulin sensitivity and immune cells for immune-mediated diseases.[54] This suggests that $p_{12}$ should depend explicitly on the specific pair of traits under consideration, including cell type in the case of eQTL or chromatin mark studies. One avenue for future exploration is whether fold change in enrichment of open chromatin/GWAS signal overlap between cell types could be used to modulate $p_{12}$ and select larger values for more *a priori* relevant tissues.

The other focus of this paper is on dealing with multiple causal variants for single traits in a single region. Single coloc can be misleading when there are completely shared causal variants in the two traits, but with different effect sizes, such that colocalisation concludes there are

single effects in each trait, different to each other (e.g. row 5 of Fig 8). Inference is much improved with conditioning, and we hope that by including the conditioning method within coloc we will enable more widespread use of this step. Note that if the two traits are measured in different populations, then colocalisation can still be performed, with a separate LD matrix for each. However, if the summary statistics from a single trait are the results of meta analysis of different populations, then conditioning needs to be performed in each population separately.

One advantage of coloc has been the minimal amount of data pre-processing required. In particular, there is no need to harmonize alleles between the two datasets or to some reference dataset. However, harmonization cannot be avoided if multiple causal variants are to be dealt with via conditioning. Here, we propose successively masking most associated SNPs and SNPs in LD with them. This has conceptual similarities to clumping, used in polygenic risk score construction to select the strongest signal in each LD-independent set of SNPs [56], and to the division of the genome into LD-independent blocks, [57] but differs to each. Our motivation is inverted compared to that for clumping: We aim to identify the set of SNPs whose GWAS summary statistics are likely to be unrelated to the masked signal, rather than select a single SNP from the masked group. We also select smaller sets of SNPs than found by dividing the genome into blocks, because we select SNPs only according to LD with the *sentinel SNP*, rather than finding breakpoints such that *every SNP* in a block is likely to have minimal LD with any SNP outside that block. While masking loses accuracy in comparison to conditioning, it improves on single coloc, and importantly doesn't appear to lead to erroneous positive conclusions for $H_4$ when $H_3$ is true, although the reverse—supporting $H_3$ for a secondary comparison when $H_4$ is true—can occur when causal variants are themselves in LD. Therefore secondary $H_3$ conclusions should be treated with some caution, but secondary $H_4$ conclusions may signal true colocalisations that would have otherwise been missed. Often a researcher may be colocalising results from one dataset for which they have complete information (e.g. because it was generated in their lab) with a public disease GWAS with less information, and here we recommend the hybrid strategy of conditioning in the dataset with full information and masking in the public dataset. Masking is also likely to avoid substantial errors in the results of approximate conditioning that can occasionally result from small deviations from LD estimated in a reference population to that in the study sample, particularly when the reference population is smaller than that used to the GWAS [58].

While we have discussed the thought process required to consider prior parameter values, thought is also required to interpret partially colocalising signals (i.e. a convincing mixture of one colocalising and one non-colocalising variant). When the two datasets are different disease GWAS, it may be reasonable that they share only one signal, with the alternate signal operating through a different mechanism. But if there are two signals for an eQTL only one of which colocalises with a disease signal, then this should be interpreted with greater caution than complete colocalisation. It suggests that there are two ways of modifying expression of a gene but that only one of those ways is also associated with variable disease risk. This might mean that the right gene has been identified in the wrong tissue, given the overlap in eQTL signals between tissues, [45] but it might also indicate incidental colocalisation. Similarly, lack of colocalisation may indicate only that the correct tissue or state has not been assayed. We anticipate that systematic analysis of multiple tissues and genes with a single disease may lead to a set of posterior probabilities that are jointly more amenable to interpretion than a single isolated analysis. However, colocalisation will always be limited by its basis in analysis of observational data, and experimental manipulation through CRISPR or through genotype-targeted assays will be required to establish causality.

In summary, we find that coloc default values for the prior probabilities of single trait association, $p_1$, $p_2$, are well supported by data across a range of data types, but that the choice of $p_{12}$ needs careful thought, and is expected to vary according to the pair of traits being considered. We recommend taking some time to do this before any analysis, documenting and justifying choices, using the coloc explorer app to translate between per-SNP and per-hypothesis values. The simulations here (Fig (4)) suggest that $p_{12} = 5 \times 10^{-5}$ provides a reasonable balance between power and false positive calls, but it is unlikely that any single point distribution on $p_{12}$ captures all prior knowledge. As varying $p_{12}$ can sometimes have a substantial impact on inference, we strongly advise users to perform sensitivity analysis for key results. Both the justification of choices and the results of sensitivity analyses should be presented to accompany any published results.

## Materials and methods

Code to run the simulations and analyses described below is available at https://github.com/chr1swallace/coloc-mask-paper.

A statistical description of the coloc method, including calculation of per-SNP and per-hypothesis Bayes factors and posterior probabilities is given in S1 Text.

To calculate the posterior probability of association shown in Fig 3c and 3d, we use the Bayes factor for association at a single SNP defined in S1 Text, $BF_1$. We calculate the posterior probability for association as a function of the prior probability that a SNP is associated with the trait, $\pi$, as

$$
\begin{aligned}
P(J_1|\text{Data}) &= \frac{P(\text{Data}|J_1)\pi}{P(\text{Data}|J_1)\pi + P(\text{Data}|J_0)(1 - \pi_0)} \\
&= \frac{BF_1\pi}{BF_1\pi + (1 - \pi_0)}
\end{aligned}
$$

where we use $J_0$, $J_1$ to denote the competing hypotheses of association and non-association at this SNP.

## Simulations

We evaluated different prior parameter settings, sensitivity analysis, or strategies for dealing with multiple causal variants by simulation. In each case, we simulated GWAS data by sampling $2N$ haplotypes of length $M$ SNPs for $N$ individuals from 1000 Genomes samples (either EUR or YRI), and selected one or two causal variants at random from amongst common SNPs (MAF>5%) according to the question being addressed.

Effect estimates at each variant were sampled from the set {0.17, 0.33, 0.50, 0.67, 0.83, 1.00, 1.17, 1.33, 1.50}, sample sizes $N$ from the set {100, 200, 500, 1000, 2000, 5000, 10000} and number of SNPs $M$ from {250, 500, 750}. Quantitative traits with residual standard deviation 1 were then simulated according to linear models, i.e. as

$$
Y = \sum_i b_i G_i + e
$$

where $i$ indexes causal variants, $b_i$ and $G_i$ the effect estimate and genotype at variant $i$, and $e \sim N(0, 1)$.

For all analyses, we used $p_1 = p_2 = 10^{-4}$ and varied $p_{12}$ as described in the text.

### GTEx analysis

We used GTEx data to estimate the probability that a random SNP could be causally associated with the expression of a gene within some bp-defined window. We analysed GTEx v7 Whole Blood significant eQTLs, downloaded from https://storage.googleapis.com/gtex_analysis_v7/single_tissue_eqtl_data/GTEx_Analysis_v7_eQTL.tar.gz on 25 June 2019. We used masking to define independent signals within this set for each gene ($r^2 < 0.01$) using 1000 Genomes EUR samples to estimate LD. We estimated q as the ratio of the number of significant lead eQTLs in multiples of 100 kb windows around the TSS to the number of SNPs in 1000 Genomes with SNPs grouped by MAF into 5 groups: [0, 0.1], (0.1, 0.2], (0.2, 0.3], (0.3, 0.4], (0.4, 0.5].

### GWAS catalog analysis

We used the GWAS summaries in the GWAS catalog (https://www.ebi.ac.uk/gwas/api/search/downloads/full, download date: 12 June 2019) to estimate the proportion of common SNPs that were independently associated with any given case/control or quantitative trait and examined how this varied according to reported sample size.

## Supporting information

**S1 Table. Summary of applied papers from 2018 using coloc.**
(PDF)

**S1 Text. Supporting mathematical derivations.**
(PDF)

**S1 Fig. Example of sensitivity analysis on a dataset which shows evidence for colocalisation at a predefined rule of posterior $P(H4) > 0.5$ across a wide range of $p_{12}$.**
(TIF)

**S2 Fig. Average posterior probabilities for each hypothesis when trait 1 has two causal variants, and trait 2 has just one, according to whether the maximum $r^2$ between multiple causal variants is $\leq 0.01$ or $> 0.01$.**
(TIF)

**S3 Fig. Average posterior probabilities for each hypothesis when both traits have two causal variants, according to whether the maximum $r^2$ between multiple causal variants is $\leq 0.01$ or $> 0.01$.**
(TIF)

## Acknowledgments

We thank Stasia Grinberg and members of the BSU for helpful discussions during the preparation of this manuscript.

The Genotype-Tissue Expression (GTEx) Project was supported by the Common Fund of the Office of the Director of the National Institutes of Health, and by NCI, NHGRI, NHLBI, NIDA, NIMH, and NINDS.

The NHGRI-EBI GWAS Catalog is funded by NHGRI Grant Number 2U41HG007823, and delivered by collaboration between the NHGRI, EMBL-EBI and NCBI.

## Author Contributions

**Conceptualization:** Chris Wallace.

**Formal analysis:** Chris Wallace.

**Funding acquisition:** Chris Wallace.

**Methodology:** Chris Wallace.

**Software:** Chris Wallace.

**Writing – original draft:** Chris Wallace.

**Writing – review & editing:** Chris Wallace.

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
