## [Decision Letter · Decision Letter 0]

26 Feb 2020

Dear Dr Wallace,

Thank you very much for submitting your Research Article entitled 'Eliciting priors and relaxing the single causal variant assumption in colocalisation analyses' to PLOS Genetics. Your manuscript was fully evaluated at the editorial level and by independent peer reviewers. The reviewers appreciated the attention to an important topic but identified some aspects of the manuscript that should be improved.

We therefore ask you to modify the manuscript according to the review recommendations before we can consider your manuscript for acceptance. Your revisions should address the specific points made by each reviewer.

[LINK]

Yours sincerely,

Michael P. Epstein

Associate Editor

PLOS Genetics

Hua Tang

Section Editor: Natural Variation

PLOS Genetics

Reviewer's Responses to Questions

**Comments to the Authors:**

Reviewer #1: Colocalization is an increasingly important aspect of genetic fine mapping efforts (>60 papers in 2018) but, unusually in statistical genetics, the most popular software (“coloc”) implements a Bayesian analysis with subjective priors. This paper demonstrates the potential sensitivity of coloc to the prior probability of colocalization, examines a huge amount of data to elicit suggestions for setting reasonable values, and provides software for performing sensitivity analysis. In addition, the assumption of one causal variant per region per trait is examined and a new approach, called masking, is suggested for situations in which current methods cannot be applied. Overall this paper gives useful guidance to users of coloc and provides insights into the method that should be of value.

Minor comments

1. P4 L52 “ubiquity of genetic effects … concordant with an omnigenic model” – suggests that such ubiquity has been established when it remains a conjecture. Some rewording needed.

2. The Introduction starts off by introducing MR and appears to motivate colocalization primarily as a way to validate instruments in MR studies. But, as seen elsewhere in the paper, most of the applications are in delineating molecular pathways to disease. I’d suggest reworking the opening paragraph to better reflect the broader motivations for colocalization.

3. P7 L101 full text could be accessed for only 25 of 60 papers. Was this due to limitations of institutional subscriptions? Could not the corresponding authors provide manuscripts for research purposes?

4. P7 L104 it would be interesting to know also how many papers used eCaviar or some other method to deal with multiple causal variants. Also, how often did the original discovery studies perform conditional analyses and rule out additional causal variants? So that when going to colocalization, the single causal variant assumption can be justified to some extent.

5. P7 L107 “prior probability … will depend” – should say “may depend” since at this point we haven’t established this, and anyway since priors are subjective the user is free to believe that there is no dependence on the traits (but may then draw the wrong conclusion).

6. P8 L124 “more likely” should be “relatively more likely”, otherwise this sentence is confusing. Initially I found this sentence counter-intuitive – seems that by looking at fewer SNPs we are more likely to find colocalization – but the point is that the prior probability of colocalization is higher relative to distinct variants when fewer SNPs are considered. However the lower number of SNPs would provide less evidence for colocalization so this is a false economy. Anyway some interpretation should be added to this and the previous paragraph as it is unclear what one should conclude from the observations.

7. P8 L132 note that all the estimates of p’s and q’s are based on statistically significant SNPs, and the number of truly associated variants must be larger. So the elicited priors must be lower bounds. What implications does this have for the final inferences?

8. P9 L145 not clear how to get a posterior probability of association from just a prior and a p-value.

9. P10 L163, 165 the Appendix was not available to review.

10. P12 L208 “unlinked” -> “not in linkage disequilibrium”. There is a difference between linkage and LD.

11. P12 The masking method still needs an LD matrix, so the only real advantage over CoJo is that there is no need to align the alleles.

12. P12 The masking method looks a lot like “clumping” as often used, for example, in constructing polygenic risk scores. Please clarify the difference, or use the same term to prevent jargon creep.

13. P213 Figure 6 caption, “setting to 1 the Bayes factor” –the main text suggests setting the log Bayes factor to -3. Log in what base?

14. P14 L244 is it feasible to make the sensitivity analysis a default action in coloc, with the results being returned in the same object as the posteriors?

15. P16 the Discussion would benefit from a summary take-home message, such as that the default values of p1 and p2 are OK but p12 needs more thought (and a summary of how to do this would also help).

Typos etc

1. P3 L32 “underly” -> “underlie”

2. P4 L59 “For example…” – the sentence has no active verb.

3. P7 L117 delete “a”; change final “,” to “.”

4. P8 L140 the double “-“ is confusing, suggest just saying “to” or writing as an interval.

5. P9 L150 “One” -> “On”

6. P11 L178 in the equation below, can delete the intersection with A2 in the third expression.

7. P11 L180 spelling of “asymmetric”

8. P12 Figure 5 caption line 2, “belief” -> “beliefs”. What does the dotted line marked “results” mean?

9. P12 L212 “is” -> “are”

10. P14 L234 “are” -> “is”

11. P16 L288 “interpretable” -> “interpretation”

12. References are a bit sloppy, eg page numbers for refs 11 and 14.

13. Supplement P1, footnote 4 “P=1105” etc looks incorrect.

Reviewer #2: This paper considers two important extensions to the currently most popular

and influential colocalization method/software "coloc": a more suitable

prior specification (than the current default) and relaxing the assumption

of only one causal SNP. In particular, the first problem has been largely

ignored in practice while its implication is significant, as the

author has clearly shown in the paper. Although the proposed methods are

not technically sophisticated, they can be tremendously useful as implemented

in the "coloc" software. The paper was well written. I only have two very

minor comments.

Minor comments:

1. Prior elicitation is a well known and general problem in Bayesian statistics,

both important and challenging. I agree with the author on all her points,

and commend the author for providing a useful online tool "coloc explorer".

However, without a "default" prior, I am not sure how useful it would be to

a "typical" biologist without deep understanding of Bayesian statistics or

"coloc" method; in fact, I would be a bit worried that someone might do

"prior mining" to try to get more significant results. Some comments or

guidelines might be helpful to a typical user.

2. I completely agree with the author on both the advantages and limitations

of the conditioning approach as compared to the proposed "masking" approach.

However, if I understand correctly, with a typical small genomic region of

interest, one would potentially mask out ALL SNPs in the region that are in

LD with the lead SNP; in other words, is the new assumption simply

that there is at most only one causal SNP in EACH LD block? If true, it is

still like doing coloc analysis under the single causal SNP assumption for

each LD block, which can be too restrictive given that there are only about

two thousand (approximately independent) LD blocks in the huiman genome.

Some clarifications and comments would be helpful.

Reviewer #3: This manuscript investigated how to derive data driven priors for best power of COLOC, provided a sensitivity analysis framework to assess the robustness of priors, and proposed a new masking approach for dealing with scenarios with multiple signals per region. It is very useful to provide guidelines for users of COLOC about how to setup priors to achieve the best power. However, this paper does not provide a clear guideline to readers. I have the following comments:

1) It would help refresh reader’s mind if a brief description about the statistical procedure of the COLOC tool could be provided either in the Introduction section along with the five stated hypotheses, or at the beginning of the Results section.

2) I think it would be helpful to make a clear guideline table for readers, e.g., suggestive p1, p2, p12 prior values for a few different combinations of number of SNPs in the test region, total number of trait signals, if multiple signals exist in the test region. Or a such table could be provided for GTEx expression traits of different tissue types, which will provide readers a concrete example.

3) It would be helpful if the authors could provide some descriptions about “coloc explorer” and “condmask coloc” and how to implement these two tools in the supplementary text.

**Have all data underlying the figures and results presented in the manuscript been provided?**

Reviewer #1: Yes

Reviewer #2: Yes

Reviewer #3: Yes

PLOS authors have the option to publish the peer review history of their article (what does this mean?). If published, this will include your full peer review and any attached files.

Reviewer #1: No

Reviewer #2: No

Reviewer #3: No

---

## [Editor Report · Decision Letter 1]

17 Mar 2020

Dear Dr Wallace,

We are pleased to inform you that your manuscript entitled "Eliciting priors and relaxing the single causal variant assumption in colocalisation analyses" has been editorially accepted for publication in PLOS Genetics. Congratulations!

Yours sincerely,

Michael P. Epstein

Associate Editor

PLOS Genetics

Hua Tang

Section Editor: Natural Variation

PLOS Genetics

Comments from the reviewers (if applicable):

**Data Deposition**

http://datadryad.org/submit?journalID=pgenetics&manu=PGENETICS-D-19-02090R1

**Press Queries**

---

## [Editor Report · Acceptance letter]

9 Apr 2020

PGENETICS-D-19-02090R1 

Eliciting priors and relaxing the single causal variant assumption in colocalisation analyses 

Dear Dr Wallace, 

We are pleased to inform you that your manuscript entitled "Eliciting priors and relaxing the single causal variant assumption in colocalisation analyses" has been formally accepted for publication in PLOS Genetics! Your manuscript is now with our production department and you will be notified of the publication date in due course.

With kind regards,

Kaitlin Butler

PLOS Genetics

On behalf of:
